# Antiviral Activities of Officinaloside C against Herpes Simplex Virus-1

**DOI:** 10.3390/molecules27113365

**Published:** 2022-05-24

**Authors:** Ji Xiao, Miaomiao Cai, Yifei Wang, Ping Ding

**Affiliations:** 1School of Pharmaceutical Sciences, Guangzhou University of Chinese Medicine, Guangzhou 510405, China; jixiao_1992@163.com (J.X.); caimiaomiao1412@126.com (M.C.); 2Guangzhou Jinan Biomedicine Research and Development Center, College of Life Science and Technology, Jinan University, Guangzhou 510632, China

**Keywords:** *Morinda officinalis*, iridoids, herpes simplex virus-1, antiviral activity

## Abstract

The iridoid compounds in traditional Chinese medicine play a prominent role in their antiviral effects. We previously reported the anti-inflammatory effect of new iridoids from the aerial parts of *Morinda officinalis*. Nevertheless, several open questions remain to explore the other biological functions of these new iridoid compounds. Herpes simplex virus-1 (HSV-1) is one of the most prevalent pathogens in human beings worldwide and due to limited therapies, mainly with the guanosine analog aciclovir (ACV) and other analogs, the search for new drugs with different modes of action and low toxicity becomes particularly urgent for public health. This study aimed to explore the anti-HSV-1 effects of iridoids from the aerial parts of *Morinda officinalis*. The dried aerial parts of *Morinda officinalis* were extracted with 95% ethanol and systematic separation and purification were then carried out by modern column chromatography methods such as silica gel column, RP-ODS column, Sephadex LH-20 gel column, and semi-preparative liquid phase, and the structure of these compounds were identified through the physical and chemical properties and a variety of spectral techniques. The obtained seven new iridoid compounds were screened for antiviral activity on HSV-1 through CCK8 and the cytopathic effect, and then the plaque reduction assay, the anti-fluorescence reporter virus strain replication, and RT-qPCR experiments were carried out to further evaluate the antiviral effect. Seven new iridoid compounds (officinaloside A–G) were identified from the aerial parts of *Morinda officinalis*, and officinaloside C showed anti-HSV-1 activity. Further functional experiments confirmed that officinaloside C has a significant inhibiting effect on HSV-1 virus plaque formation, viral gene, and protein expression, and fluorescent virus replication. Our findings suggest that officinaloside C has significant inhibitory effects on viral plaque formation, genome replication, and viral protein expression of HSV-1 which implies that officinaloside C exhibits viral activity and may be a promising treatment for HSV-1 infection.

## 1. Introduction

In recent decades, large-scale outbreaks of infectious diseases caused by COVID-19, SARS, avian influenza, Zika virus, etc. have seriously endangered human life and health. One of the most common human pathogens, herpes simplex virus type I (HSV-1), infects more than two-thirds of the world’s population [1]. HSV-1 mainly infects the central nervous system, mouth, lips, skin, and mucous membranes [2,3]. After infection, the virus can spread anteriorly or retrogradely between nerve cells, which often causes various diseases [4]. For example, severe skin infections, herpetic encephalitis, fatal encephalitis, and meningitis cause great pain and injury to the body if these infections occur repeatedly. Moreover, the incidence of neurogenic encephalitis can reach 70% without effective treatment [5]. Currently, nucleoside drugs such as aciclovir, ganciclovir, and famciclovir are clinically used for the treatment of HSV-1 [6]. Since most of these drugs have medication unicity and mutagenicity, resistance will appear after long-term treatment [7,8]. Therefore, it is urgent to develop new drugs with good antiviral activity and low toxicity simultaneously.

An array of studies has reported that iridoid compounds from traditional Chinese medicine showed prominent anti-viral effects. For instance, two new iridoid glycosides 6-*O*-trans-*p*-coumaroyl-8-*O*-acetylshanzhiside methyl ester and its cis isomer from *Barleria priorities* showed potent antiviral activity against the respiratory syncytial virus [9]; iridoid isomers lamiridosins A and B from the flowering tops of *Lamium album* were found to significantly inhibit hepatitis C virus entry in vitro [10]; asperulosidic acid from the fruit of *Morinda citrifolia* could exhibit Epstein-Barr virus early antigen activation [11]. The abundant presence of iridoids in the morinda genus hints to us that these iridoids from the aerial parts of *Morinda officinalis* (MO) could also possess antiviral activity. The root of MO (Rubiaceae), commonly used as medicinal herbs for tonifying kidney yang, strengthening the bones and muscles, and dispelling wind dampness to alleviate impotence, menstrual disorders, osteoporosis, and rheumatoid arthritis, is widely distributed in southeastern China [12,13]. Various types of secondary metabolites have been identified in MO, including iridoids, anthraquinones, polysaccharides, oligosaccharides, and triterpenoids [14]. Previously we have isolated 16 iridoid compounds from the aerial parts of MO and elucidated their anti-inflammatory mechanisms [15]; herein, studies were carried out to investigate their anti-HSV-1 activity to maximize the utilization and development of the aerial parts of MO. 

## 2. Results

### 2.1. Chemical Compounds from MO 

The dried and chopped aerial parts of MO (10.0 kg) were extracted and then refluxed with 95% EtOH to yield a crude extract (410.1 g). The extract was suspended in water and then partitioned with petroleum ether, ethyl acetate, and *n*-BuOH (each 3 × 1 L). The *n*-BuOH layer (65.9 g) was fractionated by silica gel CC to obtain six fractions (A–F). Fraction B was repeatedly chromatographed on different columns such as silica gel CC, RP-C_18_, Sephadex LH-20 CC, and semi-preparative HPLC, to yield officinaloside E (3.0 mg, t_R_ = 6.0 min), officinaloside A (7.0 mg, t_R_ = 4.9 min), officinaloside C (4.2 mg), officinaloside B (21.9 mg, t_R_ = 7.8 min), and officinaloside D (4.2 mg, t_R_ = 5.1 min). Similarly, officinaloside F (71.0 mg) and officinaloside G (56.0 mg) were obtained from fraction D. The chemical structures of compounds officinaloside A–G were mainly identified by nuclear magnetic resonance, mass spectrum, UV spectroscopy, and infrared absorption spectroscopy. The specific data shown in our previous article [15] and their structures are in Figure 1.

### 2.2. Antiviral Activity and Cytotoxicity Evaluation of Officinaloside C 

To determine whether these iridoid compounds have anti-HSV-1 activity, we screened their cytopathic effect (CPE) and the results are shown in Table 1. The data indicated that officinaloside C showed potent antiviral activity at a concentration of 10 μM. To assess the toxicity of officinaloside C, a CCK8 assay was executed to determine the CC_50_ (50% cell survival rate). The results suggested that the CC_50_ of officinaloside C was higher than 100 μM, both on the Vero and SH-SY5Y cell lines (Figure 2A,B). Above all, these results hinted that officinaloside C has weak cytotoxicity, accompanied by a strong antiviral activity.

### 2.3. Officinaloside C Can Inhibit Virus Plaque Formation

The plaque test is a classic indicator in virology research, which can quantitatively reflect the degree of viral infection. To explore the effect of officinaloside C on the replication of HSV-1 in Vero cell, we conducted a plaque reduction experiment by adding 30 PFU/well of virus dilution and different concentrations of drugs (0, 6.25, 12.5, 25, and 50 μM), each at a volume of 100 μL, to infect the cells. The results showed that the number of plaques was the largest in the control group without drug intervention, and the number of plaques gradually decreased with the increase of the administered concentration, indicating that officinaloside C can be reduced in a concentration-dependent manner (Figure 3A). There is a significant difference in the number of plaques infected by the virus at 25 μM, which preliminarily proves the antiviral activity of the compound.

### 2.4. Officinaloside C Has Inhibitory Effects on Gene Expression after HSV-1 Infection

In order to detect the effect of officinaloside C on HSV-1’s immediate early gene, early gene, and late gene expression, the representative genes *UL54*, *UL52,* and *UL27* of each period were selected, respectively. According to the characteristics of the HSV-1 immediate early gene *UL54*, early gene *UL52*, and late gene *UL27* expression changes [16], changes in the expression levels of *UL54* at 3 h after infection, *UL52* at 6 h after infection, and *UL27* at 9 h after infection, are detected. The results are shown in Figure 3B below. Officinaloside C has a significant inhibitory effect on the expression of *UL54*, *UL52,* and *UL27* genes at 25 μM, confirming the antiviral activity of officinaloside C on HSV-1.

### 2.5. Officinaloside C Can Inhibit Virus Replication

Fluorescence microscopy results showed that, after 48 h of infection with HSV-1 green fluorescent protein (GFP), without drug intervention, the GFP fluorescence signal of HSV-1 was very strong and widely distributed in the field of view. In the intervention of different concentrations of drugs, the fluorescence signal of HSV-1 GFP was significantly weakened, and the fluorescence signal decreased with the increase of the administered concentration. When the administered concentration was 50 μM, a significant difference was observed compared with the control group. The antiviral activity of officinaloside C is shown in Figure 3C. 

### 2.6. Officinaloside C WB Results Analysis

Next, we further evaluated the effect of officinaloside C on viral protein expression in the HSV-1 F strain. Vero cells were infected with HSV-1 at MOI = 1 in the presence of increasing concentrations of officinaloside C from 6.25–50 μM for 24 h. Then, viral glycoprotein B (gB) was detected by Western blot assay. As shown in Figure 3D, officinaloside C significantly inhibited gB expression at 6.25–50 μM concentrations. HSV-1 entry needs four glycoproteins: -gB, gH, gL, and gD-. [17]. Therefore, it is possible that the inhibitory effect of officinaloside C on viral replication is due to the inhibition of gB expression.

## 3. Materials and Methods

### 3.1. Plan Material and Isolation of New Iridoid Compounds of Officinaloside A–G

The aerial parts of *Morinda officinalis* were collected in July and August 2019, from Guangdong China. The plant material was identified by Professor Ping Ding of Guangzhou University of Chinese Medicine. A voucher specimen (No. 20190731001) was deposited in the School of Pharmaceutical Science, Guangzhou University of Chinese Medicine. New iridoid compounds of officinaloside A–G were isolated from the aerial parts of MO as previously described [15].

### 3.2. Cells and Viruses

Vero cell line (ATCC, Manassas, VA, USA) was cultured in Dulbecco’s modified Eagle’s medium (DMEM; 8118305, GIBCO/Thermo Fisher Scientific, Gaithersburg, MD, USA) with 10% fetal bovine serum (11011-8611, TIANHANG, Hangzhou, China). SH-SY5Y (ATCC, Manassas, VA, USA), a human neuroblastoma cell line was cultured in Dulbecco’s modified Eagle’s medium (DMEM; 8118305, GIBCO/Thermo Fisher Scientific, Gaithersburg, MD, USA) with 10% fetal bovine serum (FND500, ExCell Bio, Shanghai, China). HSV-1 strain F (ATCC, Manassas, VA, USA), initially obtained from Hong Kong University, was propagated in Vero cells and stored at −80 °C until use. EGFP-HSV-1, expressing an EGFP-tagged viral protein Us11, a kind gift from the College of Pharmacy, Jinan University (Guangzhou, China), was used to evaluate viral replication capacity [18]. 

### 3.3. Cytotoxicity Assay

Vero cells (1 × 10^5^ cells/well) were seeded in 96-well plates and cultured overnight in a humidified cell culture incubator with 5% CO_2_ at 37 °C. The supernatant was then removed and new medium with iridoid solutions (100 μM, 50 μM, 25 μM, 12.5 μM, 6.25 μM, 3.12 μM, 1.56 μM, 0.781 μM, 0.391 μM, and 0 μM, respectively) were added. The cytotoxicity of iridoid compounds was measured by CCK8 assay (Glpbiotechnology, Montclair, CA, USA). After 48 h, the supernatant was removed and 100 µL of the medium with 10% CCK8 reagent was added to each well and incubated for 2 h. The OD value was recorded by a microplate reader at 450 nm. Cell viability (%) = (OD_drug group_ − OD_blank group_)/(OD_control group_ − OD_blank group_) × 100%. The 50% cytotoxicity concentration (CC_50_) was defined as the concentration required to reduce cell viability by 50%.

### 3.4. Antiviral Assay

The antiviral activity of the new compounds against HSV-1 was first evaluated by a cytopathic effect reduction assay and later confirmed by a plaque reduced assay. For the plaque reduced assay, Vero cells (1 × 10^5^ cells/well) were plated in 24-well plates and when the wells contained confluent monolayers, they were then infected with HSV-1 suspensions to produce 20–30 plaques per well. After 2 h incubation with 5% CO_2_ at 37 °C, unabsorbed virions were aspirated and washed twice with PBS. Next, iridoid solutions (50 μM, 25 μM, 12.5 μM, 6.25 μM and 0 μM, respectively), and ACV (20 μM) were added to the wells and DMSO (0.1%) was set as a control group. These plates were put into the 5% CO_2_, 37 °C environments for another 2 h, right after 1% methylcellulose in the nutrient medium was added. After 72 h of incubation, the cells were fixed with 4% paraformaldehyde and stained with 1% crystal violet for 20 min. Finally, the number of plaques was counted per well, respectively. The percentage of viral inhibition (%VI) was calculated as %VI = 1 − (PFU in treated cells/PFU in control cells)] × 100%, where PFU is a plaque-forming unit [16]. 

### 3.5. Fluorescence Analysis

Vero cells (2 × 10^5^ cells/well) were seeded in 6-well plates and incubated at 37 °C, 5% CO_2_ overnight. Cells infected with EGFP-HSV-1 (MOI = 0.2) were treated with iridoid solutions (50 μM, 10 μM, 2 μM, 0.4 μM and 0 μM, respectively). After 48 h incubation at 37 °C and 5% CO_2_, each well was observed by fluorescence microscopy and the fluorescent signal intensity was quantified by image analysis software (Image J, National Institutes of Health, Bethesda, MD, USA).

### 3.6. Western Blot

Vero cells were infected with HSV-1 at a multiplicity of infections of HSV-1 and treated with officinaloside C (0.4 μM, 2 μM, 10 μM, 50 μM) simultaneously. After 24 h, cells were collected and lysed in RIPA Lysis Buffer (Beyotime, Shanghai, China) containing 2% PMSF and 1% phosphatase inhibitor, and then samples were separated by 10% gradient SDS-PAGE. Next, samples were transferred to nitrocellulose and incubated with primary antibodies overnight at 4 °C, followed by incubation with appropriate secondary antibodies (Beyotime, Shanghai, China) (1:6000–8000 dilution) for 60 min at room temperature. Target proteins were detected by enhanced chemiluminescence, and GAPDH was used as an internal standard.

### 3.7. Real-Time Fluorescent Quantitative PCR (RT-PCR)

SH-SY5Y cells infected with HSV-1 (MOI = 1) were treated with iridoid compounds. At 3, 6, and 9 h post-infection (h p.i.), total RNAs were extracted with TRIzol reagent (Beyotime, Shanghai, China), and the cDNA was obtained by reverse transcription (PrimeScript RT reagent kit, Takara, Kyoto, Japan). Then, qPCR was performed on the cDNA products using an SYBR Green PCR Premix *Pro Taq* HS qPCR Kit (Accurate biology, Hunan, China), following the manufacturer’s protocol. The expressions of *UL54*, *UL52,* and *UL27* were quantified and analyzed by a BioRad CFX96 real-time PCR system, and GAPDH was used as an internal standard. The primer sequences were as follows: HSV-1 *UL54* F (5′-TGGCGGACATTAAGGACATTG-3′), *UL54* R (5′-TGGCCGTCAACTCGCAGA-3′), *UL52* F (5′-GACCGACGGGTGCGTTATT-3′), *UL52* R (5′-GAAGGAGTCGCCATTTAGCC-3′), *UL27* F (5′-GCCTTCTTCGCCTTTCGC-3′), *UL27* R (5′-CGCTCGTGCCCTTCTTCTT-3′), *GAPDH* F (5′-GTCATTGAGAGCAATGCCAG-3′), *GAPDH* R (5′-GTGTTCCTACCCCCAATGTG-3′).

### 3.8. Statistical Analysis

Data were expressed as the mean ± SD for triplicate independent experiments. Statistically significant values were compared using one-way ANOVA with Dunnett’s test by GraphPad Prism 8.0, and *p*-values of less than 0.05 were statistically significant.

## 4. Discussion

Traditional Chinese medicine (TCM) plays a pivotal role in the research and development of new drugs, which highlights the importance of the inheritance, innovation, and development of TCM. Globally, the fight against the COVID-19 pandemic not only made the world aware of the value of TCM but also accelerated the pace of introducing TCM to the world. As one of the four major southern TCM medicines, MO possesses a variety of chemical components, displaying the important value in various aspects [19]. Studies have shown that iridoid compounds have a wide range of biological activities, such as anti-inflammatory, anti-viral, anti-tumor, neuroprotective, liver protective, and hypoglycaemic effects [20,21,22], which illustrated the great potential for drug development. Meanwhile, herpes simplex viruses (HSVs) are distributed with a seroprevalence ranging up to 95% in the adult population, and herpes virus encephalitis (HSE) caused by HSV-1 infection is also increasing year by year [3,4]. What is more, according to the World Health Organization (WHO), 65% of the world’s population is using plants and herbs as part of a treatment for viral infections [23]. Natural products have always been a reliable source of new compounds with antiviral properties. Many studies have been conducted since 1995 to isolate biologically active anti-herpes-simplex-virus compounds from plants and functional foods, including *peganum harmala, pistacia vera,* and *quercus brantii* [24]. As an optic nerve virus, HSV-1 infects brain tissue through the eye or nasal cavity and causes herpesvirus encephalitis by extending the trigeminal ganglion [2]. SH-SY5Y is a human-derived cell of neural tissue origin. In this experiment, seven new iridoid compounds were screened for antiviral activity, and it was found that officinaloside C has good anti-HSV-1 activity on SH-SY5Y cells [25]. This helps to understand the virulence of officinaloside C in anti-HSV-1-infection, especially herpes encephalitis caused by anti-HSV-1-infection. Combining the results of CPE and cytotoxicity, we chose a concentration interval from 0.4 μM to 50 μM to assay the antiviral activity of officinaloside C. The results showed that officinaloside C exhibited significant inhibition of HSV-1 replication in this concentration range. However, relative to the dosing concentration of the anti-HSV-1 clinical drug ACV [26], there may not be a higher inhibition rate under the same concentration conditions again. Viral gene and protein expression is an important part of the viral life cycle, and inhibition of this process can effectively inhibit viral replication. Both the plaque reduction experiment and the fluorescent reporter gene experiment showed that officinaloside C had antiviral activity and inhibited virus replication in a concentration-dependent manner; the RT-qPCR experiment results showed that officinaloside C could inhibit the expression of HSV-1 immediate early genes, early genes, and late genes, and exert antiviral activity. Especially with the development of advanced science and technology, the continuous improvement of extraction and separation technology has been promoted, and more compounds with new structures have been continuously separated and identified [27]. This not only provides a wealth of lead compounds for the development of new drugs but has also laid a sufficient scientific basis for the discovery of more new pharmacological effects. In addition, this study only explored the isolation and antiviral activity of officinaloside C, and further study is needed on the exact molecular mechanism of the antiviral effect of officinaloside C in vitro and in vivo. 

## 5. Conclusions

To sum up, we extracted and isolated a series of compounds from the traditional Chinese herb *Morinda officinalis* and explored their chemical structures. CPE assay found that officinaloside C, one of the iridoid compounds, showed antiviral activity. Further exploration revealed that officinaloside C was able to inhibit the expression of viral genes and proteins. Our findings demonstrated that officinaloside C is a promising candidate for pharmacological therapeutics and may play a role in preventing and treating HSV-1 infection and propagation effectively.

## Figures and Tables

**Figure 1 molecules-27-03365-f001:**
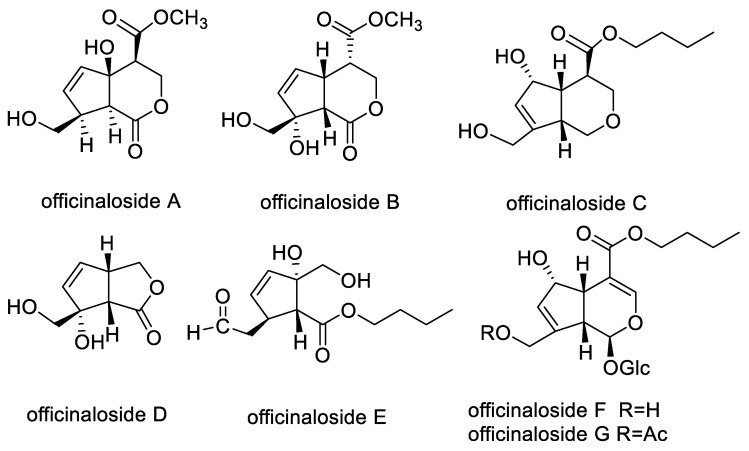
Chemical structures of officinaloside (**A**–**G**).

**Figure 2 molecules-27-03365-f002:**
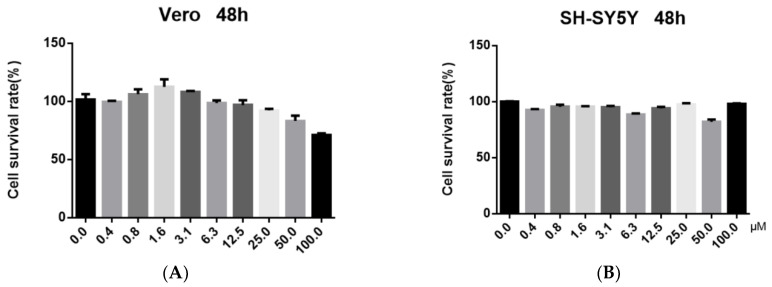
Cytotoxicity evaluation of officinaloside C on Vero (**A**) and SH-SY5Y (**B**) cell lines by using the CCK8 assay. Cell culture without iridoid compounds was used as a negative control. The data represent the percentage of cell viability in comparison with cell control, and the bars denote standard deviation (SD).

**Figure 3 molecules-27-03365-f003:**
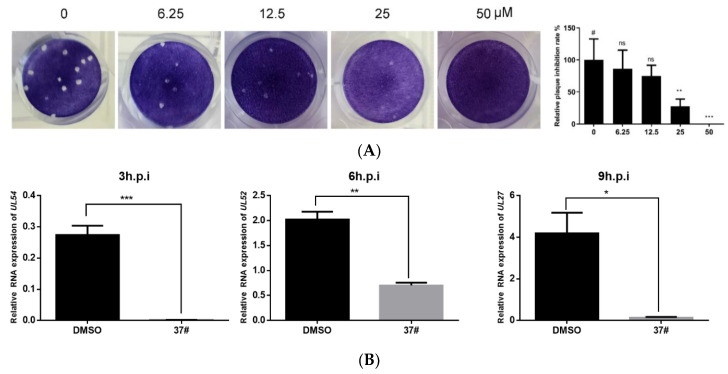
The officinaloside-C-affected virus replication and gene expression in HSV-1-infected Vero cells (*x* ± *s*, *n* =3). (**A**) Plaque reduction results showed that after adding different concentrations of officinaloside C and HSV-1 virus dilution (30 PFU/well) to Vero cells, officinaloside C can inhibit virus replication. (**B**) RT-qPCR results showed that officinaloside C has inhibitory effects on *UL54* at 3 h, *UL52* at 6 h, and *UL27* at 9 h after HSV-1 infection. (**C**) Fluorescence report test results showed that officinaloside C can inhibit viral replication. (**D**) Vero cells were treated with HSV-1 F in the presence of various concentrations of officinaloside C for 24 h, respectively. Cell lysates were then subjected to a Western blot assay to detect the protein levels of viral protein gB. GAPDH was used as a loading control (* *p* ≤ 0.05, ** *p* ≤ 0.01, *** *p* ≤ 0.001, # control group).

**Table 1 molecules-27-03365-t001:** Antiviral activity of iridoid compounds.

Compounds	100 μM	10 μM	1 μM	Conclusion
Officinaloside A	++++	++++	++++	×
Officinaloside B	++++	++++	++++	×
Officinaloside C	+	+	++	√
Officinaloside D	++++	++++	++++	×
Officinaloside E	++++	++++	++++	×
Officinaloside F	++++	++++	++++	×
Officinaloside G	++++	++++	++++	×
ACV	+	+	+	√

Note: CPE was recorded as follows: 1%~25% cytopathic lesions, “+”; 26%~50% cytopathic lesions, “++”; 51%~75% cytopathic lesions, “+++”; 76%~100% cytopathic lesions, “++++”, The “√” sign indicates antiviral activity, and the “×” sign indicates no antiviral activity.

## Data Availability

Not applicable.

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
