# Peer review of "Antiviral Activities of Officinaloside C against Herpes Simplex Virus-1"

_molecules, 2022, doi:10.3390/molecules27113365_

Round 1
Reviewer 1 Report
Submitted for review article (molecules-1649794) entitled ,, Antiviral activities of officinaloside C against Herpes simplex virus-1 ‘’is an original paper. The authors try to explain the antiviral effect of new iridoid compounds (officinaloside A-G). The article is short and in my opinion should rather be presented as short communication. Abstract and Introduction is quite clear, although in the introduction there are necessary corrections related to the formatting of the text (various fonts, grammar and punctuation errors) - please correct it. The material and methods are sufficient, although in the Western blot section, there is no information on the concentration and the compounds used in the research- please add. The results are described quite legibly, although figure numbers are missing in section 3.4-please add. Unfortunately the discussion is very laconic and in fact it does not exist. First of all, it should be separated from its ,,conclusions’’. They should be two separate parts that flow from one another. The discussion presented by the authors should be discussed primarily with the literature. This is not some new class of relationships because the authors do not present it like that, so please complete the discussion. Therefore, it is necessary to thoroughly re-arrange it. Examples and everything related to this class of relationship are missing. The literature shows quite a lot of articles on this metabolite. Moreover, the work should be meticulously checked and corrected in terms of its function, grammar and style. The authors should polish up he English language.
Author Response
Thank you very much for your suggestions. We have made the necessary corrections to the full text formatting (including various fonts, grammar and punctuation, etc.). In the Western blot section, information on the concentrations and compounds used has been supplemented. The discussion section has also been fully improved and adjusted. As well as the English language expressions have been improved.

Reviewer 2 Report
- “Morinda officinalis” change to “ officinalis” throughout the manuscript.
- Line 36: “Iridoids; Herpes Simplex Virus-1; Antivirus activity” change to “iridoids; herpes simplex virus-1; antivirus activity”
- Lines 164-165: “To determine whether the extracts………. … for antiviral activity”, the assay carried out of isolated compounds, revise.
- Line 172: “Figure 2. Cytotoxicity evaluation of iridoid compounds …….“, the cytoxicity of officinaloside C, rephrase
- Table 1: specify the symbols that follow: “++++, ++, +, ×, √“
- Line 194: “Changes” change to “changes”
- Line 194: “Figure B” change to “Figure 3B”
- Line 196: “Compound 3” change to “officinaloside C”
- Where is Figure 1, Figure 3 C,D in the text.
- Line 226: “and more”, delete
- Discussion section should be rewritten.
- Graphic is highly recommended.
- English editing is highly recommended
- The authors should follow the Journal guideline in manuscript preparation.
- Authors could benefit from the following reference:”Yosri, N., et al., (2021). Anti-viral and immunomodulatory properties of propolis: chemical diversity, pharmacological properties, preclinical and clinical applications, and in silico potential against sars-cov-2. Foods, 10(8), 1776.
Author Response
1.“Morinda officinalis” change to “ officinalis” throughout the manuscript.
Thank you for your suggestions. We have changed them throughout the manuscript.
2。Line 36: “Iridoids; Herpes Simplex Virus-1; Antivirus activity” change to “iridoids; herpes simplex virus-1; antivirus activity”
Thank you for your opinions. We have changed them.
3.Lines 164-165: “To determine whether the extracts………. … for antiviral activity”, the assay carried out of isolated compounds, revise.
Thank you. We have revised the sentence.
4.Line 172: “Figure 2. Cytotoxicity evaluation of iridoid compounds …….“, the cytotoxicity of officinaloside C, rephrase
Thank you. We have revised the sentence.
5.Table 1: specify the symbols that follow: “++++, ++, +, ×, √“
Thank you for your opinions, we have refined it.
6.Line 194: “Changes” change to “changes”
Thank you. We have revised the words and phrases.
7.Line 194: “Figure B” change to “Figure 3B”
Thank you. We have revised the words and phrases.
8.Line 196: “Compound 3” change to “officinaloside C”
Thank you. We have revised the words and phrases.
9.Where is Figure 1, Figure 3 C,D in the text.
Thank you. We have added the related words and phrases in the text.
10.Line 226: “and more”, delete
Okay. We have deleted these words.
11.Discussion section should be rewritten.
Okay. We have rewritten the discussion part.
12.Graphic is highly recommended.
Thank you for your recommendation. The grammatical problems of this paper have been modified by English masters.
13.English editing is highly recommended
Thank you for your recommendation. The article editing problems have been modified.
14.The authors should follow the Journal guideline in manuscript preparation.
Thank you very much for your suggestion. We have followed the Journal guideline in manuscript preparation.
15.Authors could benefit from the following reference:”Yosri, N., et al., (2021). Anti-viral and immunomodulatory properties of propolis: chemical diversity, pharmacological properties, preclinical and clinical applications, and in silico potential against sars-cov-2. Foods, 10(8), 1776.
Thank you very much for your suggestion. We have adjusted the format of the manuscript by referring to the literature
Reviewer 3 Report
The text bellow contains comments on manuscript entitled “Antiviral activities of officinaloside C against Herpes simplex virus-1”.
The manuscript is focused on evaluation of the antiviral activity on HSV-1 virus of several iridoid compounds from Morinda officinalis. The manuscript is well written, with logically structured experimental design, result and discussion section that logically explain the aim of the study.
I have listed some suggestions for corrections in case the authors consider them helpful:
In the introduction section the font size of the text must be unified, not to be different. I think that more focus should be paid on the formulation of the goal of the study.
Please correct all technical issues in the materials in the material and methods section, e.g please check for additional space between the words, 2 x 105 (probably should be 2x105).
Section 3.1. Chemical compounds from MO. How did you identify the compounds? Is their identification from a previous study? If yes, please mention once again in brief the analytical technique used to identify the compounds.
I think that the names of the isolated compounds should be in small caps when the names stay in the middle of the sentence.
Could you please explain in short why the neuroblastoma SH-SY5Y cell line is used to investigate anti-viral activity?
In section 4 you still have text colored in gray, please correct it.
Author Response
In the introduction section the font size of the text must be unified, not to be different. I think that more focus should be paid to the formulation of the goal of the study.
Thank you for your suggestions. We have changed the formulation problems in the article.
Please correct all technical issues in the materials in the material and methods section, e.g please check for additional space between the words, 2 x 105 (probably should be 2x105). Thank you for your suggestions. We have changed these issues throughout the article.
Section 3.1. Chemical compounds from MO. How did you identify the compounds? Is their identification from a previous study? If yes, please mention once again, in brief, the analytical technique used to identify the compounds.
Thank you for your suggestions. We have revised these questions as you recommended.
I think that the names of the isolated compounds should be in small caps when the names stay in the middle of the sentence. Thank you for your suggestions. We have revised the names of those compounds as you recommended.
Could you please explain in short why the neuroblastoma SH-SY5Y cell line is used to investigate anti-viral activity?
Thank you, type 1 Herpes simplex virus is a neurotropic virus, and SH-SY5Y, as a human neurotissue-derived cell line, is relatively helpful in alleviating the effects of reactive compounds on herpes encephalitis
In section 4 you still have text colored in gray, please correct it.
Thank you, we have revised these question as you recommended.
Round 2
Reviewer 1 Report
After rereading the manuscript, I believe that the authors have corrected it as suggested by the reviewer, however, the conclusions section should be after the discussion section-please correct.
Reviewer 2 Report
- Graphic abstract is highly recommended.
- Reference style should be adjusted based on journal guideline